# Impacts of hemispheric solar geoengineering on tropical cyclone frequency

Anthony C. Jones[1,2], James M. Haywood[1,2], Nick Dunstone[2], Kerry Emanuel[3], Matthew K. Hawcroft[1], Kevin I. Hodges[4] & Andy Jones[2]

Solar geoengineering refers to a range of proposed methods for counteracting global warming by artificially reducing sunlight at Earth's surface. The most widely known solar geoengineering proposal is stratospheric aerosol injection (SAI), which has impacts analogous to those from volcanic eruptions. Observations following major volcanic eruptions indicate that aerosol enhancements confined to a single hemisphere effectively modulate North Atlantic tropical cyclone (TC) activity in the following years. Here we investigate the effects of both single-hemisphere and global SAI scenarios on North Atlantic TC activity using the HadGEM2-ES general circulation model and various TC identification methods. We show that a robust result from all of the methods is that SAI applied to the southern hemisphere would enhance TC frequency relative to a global SAI application, and vice versa for SAI in the northern hemisphere. Our results reemphasise concerns regarding regional geoengineering and should motivate policymakers to regulate large-scale unilateral geoengineering deployments.

[1] College of Engineering Maths and Physical Sciences, University of Exeter, Laver Building, North Park Road, Exeter EX4 4QE, UK. [2] Met Office Hadley Centre, Fitzroy Road, Exeter EX1 3PB, UK. [3] Center for Global Change Science, Massachusetts Institute of Technology, 77 Massachusetts Avenue, Cambridge, MA 02139, USA. [4] Department of Meteorology, University of Reading, PO Box 243, Earley Gate, Reading RG6 6BB, UK. Correspondence and requests for materials should be addressed to A.C.J. (email: anthony.jones@metoffice.gov.uk)

In the last decade, solar geoengineering (SG) has rapidly garnered attention as a plausible method to counteract global warming[1–3]. Studies with general circulation models (GCMs) indicate that SG could effectively cool the Earth's surface, at the expense of regional climate changes[4], but these regional changes would be less severe than those in a non-geoengineered world[5]. A study has been performed[6] on the frequency and intensity of Atlantic hurricanes and associated storm surges using a multi-model analysis of Geoengineering Model Intercomparison Project (GeoMIP) scenarios G3 and G4[7], where stratospheric aerosol injection (SAI) is applied relatively uniformly to both hemispheres. However, a growing number of studies have investigated regional SG application scenarios, which could prove preferential to a global application by restricting the geospatial magnitude of the climate response or by being used to target specific climate changes[8–11]. SAI does not easily lend itself to regional impositions due to the rapid dispersion of aerosols in the stratosphere. Nevertheless, SAI could be contained or promoted in a single hemisphere due to the general poleward transport tendency of the stratospheric circulation[8,11]. Preferential aerosol injection in a single hemisphere would alter tropical sea-surface temperature (SST) gradients and displace the Inter-Tropical Convergence Zone (ITCZ) toward the opposite hemisphere as observed following the 20th century Katmai (1912) and El Chichón (1982) volcanic eruptions[11,12]. Consequentially, SAI concentrated in the northern hemisphere (NH) would likely reduce rainfall over the Sahel and vice versa for SAI in the southern hemisphere (SH)[11].

Another phenomenon related to the location of the ITCZ is North Atlantic tropical cyclone (TC) frequency[13,14]. An ITCZ displaced to the north provides optimal conditions for cyclogenesis promotion from African easterly waves (AEWs) in the hurricane main development region (MDR, defined as (5°–20°N, 15°–85°W)), which results in anomalously high TC activity[13–15]. In contrast, an ITCZ displaced to the south is associated with increased wind shear over the MDR and attenuated TC activity. TC activity was significantly attenuated following the El Chichón (1982) and Pinatubo (1991) volcanic eruptions, both of which primarily enhanced the NH aerosol burden[16,17]. Conversely, the Tambora (1815) and Agung (1963) volcanic eruptions primarily enhanced the SH aerosol burden, and were subsequently followed by periods of enhanced TC activity[17]. Additionally, GCM studies have implicated periods of high (low) NH-centric anthropogenic aerosol emissions with attenuated (enhanced) TC activity in the 20th century (Supplementary Note 1)[13], which further corroborates the relationship between asymmetric aerosol burdens and TC activity. As regional SAI applications have been proposed to specifically target, for instance, NH sea-ice concentrations[8] and would necessarily alter the inter-hemispheric aerosol gradient, it is instructive to assess the implications of global and regional SAI scenarios on North Atlantic TC activity.

In this study, we investigate North Atlantic TCs in simulations performed using the HadGEM2-ES GCM in a fully coupled atmosphere–ocean configuration[18] by directly tracking TC-like features[19], by utilising various metrics that have been developed as proxies for TC activity[13,20,21], and by employing a widely used statistical-dynamical downscaling model[22] (see Methods section). Our motivation for using different TC-identification methods is the significant disparity in projections of future TC activity in the

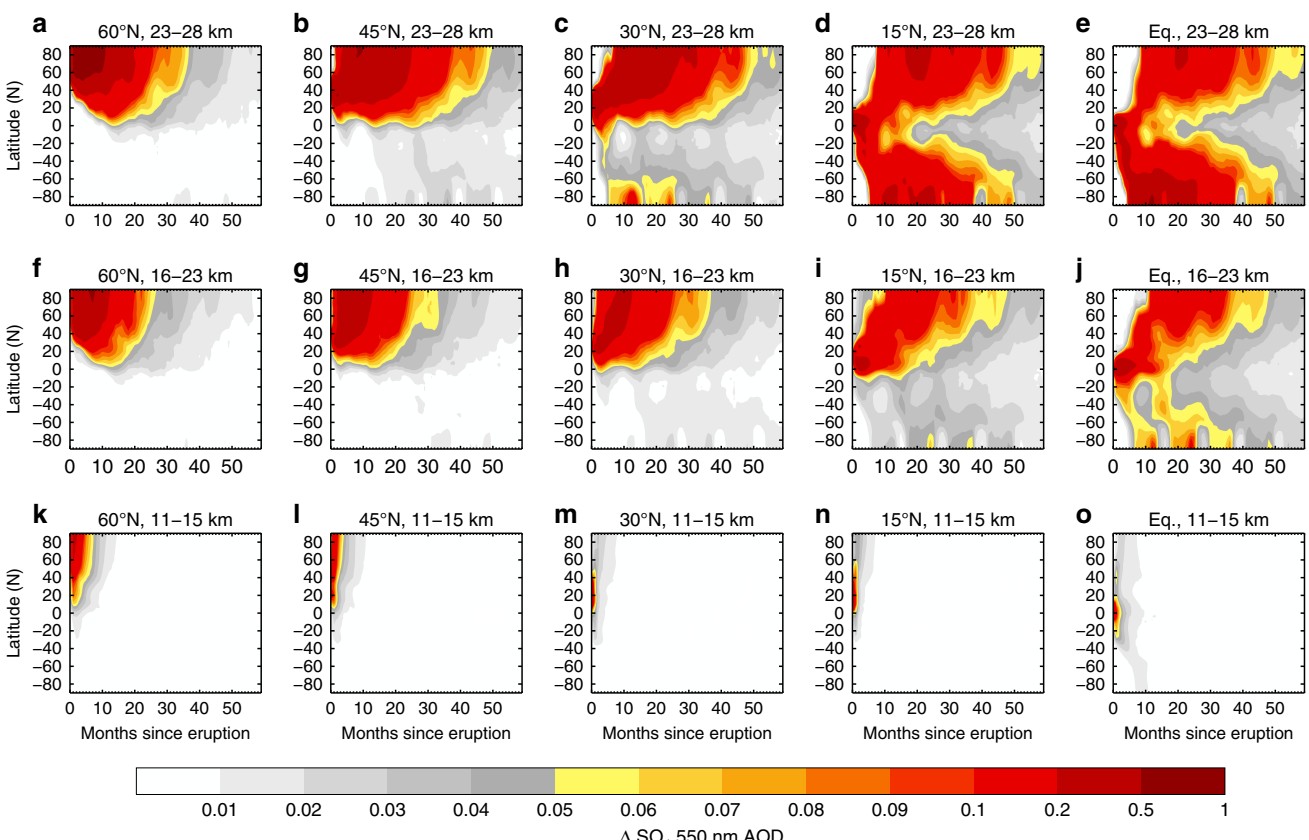

**Fig. 1** Investigating the sensitivity of aerosol dispersion to the altitude and latitude of a volcanic eruption. Five-year evolution of the anomaly in sulphate ($SO_4$) aerosol optical depth (AOD) for injections of sulphur dioxide into the Northern Hemisphere. Latitudes progress from 60°N in the left column (**a**, **f**, **k**) to the Equator in the right column (**e**, **j**, **o**) and injection altitudes from 23 to 28 km in the top row (**a–e**) to 11–15 km in the bottom row (**k–o**). Simulations are with a 'high-top' version of the HadGEM2 model with stratospheric layers up to 80 km using the CLASSIC aerosol scheme[32]

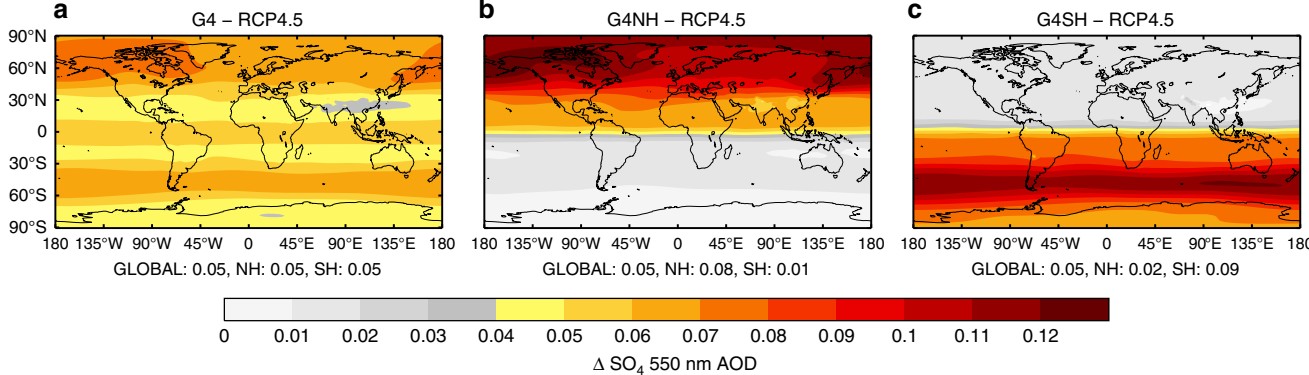

**Fig. 2** Aerosol optical depth anomalies in the solar geoengineering simulations. Sulphate ($SO_4$) 550 nm aerosol optical depth (AOD) anomaly 2020–2070 for **a**, global solar geoengineering (G4); **b**, northern hemisphere solar geoengineering (G4NH); and **c**, southern hemisphere solar geoengineering (G4SH) relative to RCP4.5

North Atlantic basin between studies with alternative algorithms[21,23,24]. In particular, the results of GCM studies mostly agree that overall TC frequency decreases under global warming, but that the frequency of the most intense storms increases[19,23,25]. Conversely, the results of applying statistical-dynamical downscaling to CMIP5 output suggest a steady increase in North Atlantic TC frequency under global warming[24]. In contrast, the results of applying a statistical relationship between TC activity and relative SSTs to CMIP5 output showed no robust trend in TC activity changes under global warming[21]. We have therefore decided to utilise all three different approaches (explicit storms, statistical relationships and downscaling) for comparison purposes and due to their relative merits (Supplementary Note 2). Note that on top of these TC identification methods, other methods have also been developed such as 'dynamical downscaling' that involves embedding a high-resolution climate model within a GCM[26], and alternative explicit TC identification algorithms such as the Camargo-Zebiak algorithm[27].

We assess two ensemble members for the recent historical period (1950–2005, hereafter denoted HIST) and three ensemble members for the Representative Concentration Pathway (RCP) 4.5 scenario (2006–2089). For SAI, we assess three ensemble members for a global SAI scenario (G4[7]) in which a constant injection rate of 5 Tg of sulphur dioxide ($SO_2$) per year is applied uniformly over the globe from 2020 to 2070, and one simulation each for NH-only (G4NH) and SH-only (G4SH) SAI scenarios in which 5 Tg[$SO_2$] per year is injected evenly over the hemisphere from 2020 to 2070[11]. We assess the impact of SAI cessation by abruptly suspending aerosol injection in year 2070 in G4/G4NH/G4SH and allowing the model to run for a further 20 years. To analyse TC frequency, we utilise ERA-interim (ERA-I) reanalyses for the period 1979–2014[28] and compare the ERA-I and simulated TC frequency with observations from the HURDAT2 Best Tracks data set[29]. We employ a widely used feature tracking software (TRACK)[19] to track vorticity maxima over the North Atlantic basin in the simulations and reanalysis data. The additional TC proxy metrics that are analysed are June–November (JJASON) precipitation in the MDR, vertical zonal-wind shear between 850 and 250 hPa in the MDR ($U_{850}$–$U_{250}$), and the difference between the SST in the MDR and the tropics as a whole (denoted relative SST)[13]. We also utilise a statistical-dynamical downscaling model to investigate changes to TC frequency and intensity[22,24]. For each of these methods, we find that TC activity is enhanced by SH-only SAI relative to a global SAI application, and vice versa for NH-only SAI. We conclude that asymmetries in the climatic response to hemispheric SG should motivate policymakers to regulate geoengineering so as to deter unilateral deployments.

## Results

**Aerosol distribution.** It is important to assess whether single-hemisphere SAI scenarios are feasible. The injection of aerosol into the stratosphere following a volcanic eruption can lead to radically different spatial and temporal distributions depending upon the altitude and latitude of the injection[30], the representation of the quasi-biennial oscillation (QBO[31]) and the local meteorological conditions that prevail at the time of the eruption[32]. We demonstrate the sensitivity to altitude and latitude by performing volcanic eruption simulations using an atmosphere only version of the HadGEM2-CCS model[32] with the model top at ~84 km. The same amount of $SO_2$ was emitted at various different altitudes and latitudes in the NH (the results from simulations emitting into the SH reveal a strong similarity and are not shown here). It is immediately evident from Fig. 1 that only injection strategies where $SO_2$ is injected into high altitudes (23–28 km) at equatorial latitudes (Equator and 15°N) lead to an aerosol distribution that is approximately hemispherically symmetric. Emissions at high altitudes (23–28 km) northward of 15°N lead to aerosol distributions that are significantly larger in the NH than the SH. Emissions at intermediate altitudes of 16–23 km altitude result in aerosol predominantly in the NH. Emissions at low altitude (11–15 km) are mostly below the tropopause leading to a much more limited lifetime of the resultant aerosol. Given current technical challenges of any deliberate SAI scheme[33], it is feasible that a sub-optimal SAI strategy (ie, at mid-latitudes and/or at lower altitudes) might conceivably be pursued.

We now consider the simulations in this study performed with the low-top version of the HadGEM2 model. As in other studies[11,31,34–36], we compensate for the lack of adequately resolved QBO owing to the limited height of the top of the model by injecting over a wide range of latitudes rather than injecting at a single point. Figure 2 shows the annual-mean sulphate ($SO_4$) aerosol optical depth (AOD) anomalies in the G4, G4NH and G4SH simulations averaged over 2020–2070. It is clear that the $SO_4$ aerosol is primarily confined to the hemisphere(s) of injection in all of the SAI scenarios.

**Climate changes.** Regional and global SAI applications would do much to ameliorate changes in near-surface air temperature and sea-ice evident in the RCP4.5 scenario, with the principal counteractive effect occurring in the hemisphere(s) of injection (Fig. 3). However, the impacts in the un-geoengineered hemisphere would also be significant owing to atmospheric and oceanic inter-hemispheric energy transport, for instance, in these simulations an NH cooling of 0.7 K is observed in G4SH relative to RCP4.5 (2020–2070), compared to 1 K in G4 and 1.1 K in

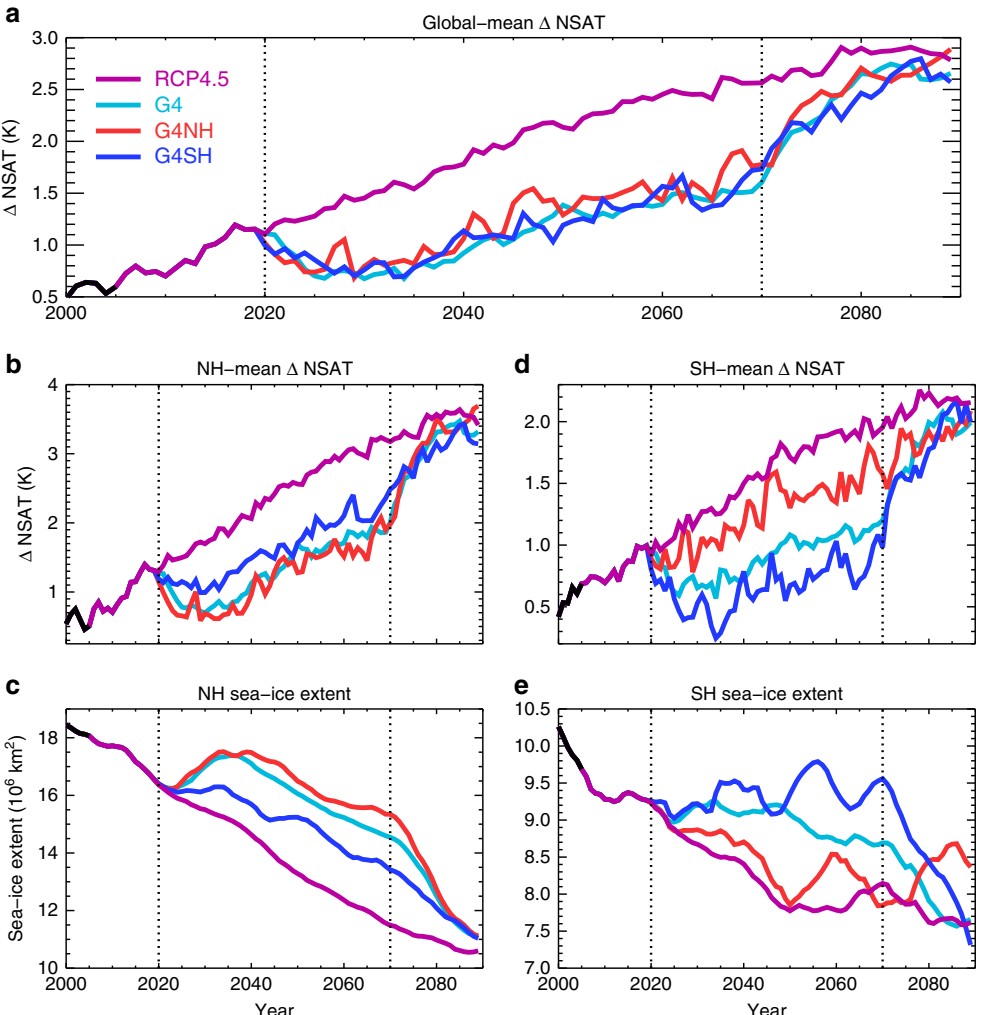

**Fig. 3** Twenty-first century temperature and sea-ice changes. **a** Global-mean near-surface air temperature (NSAT) anomaly relative to a 240-year pre-industrial control simulation for RCP4.5, global solar geoengineering (G4), northern hemisphere (NH) solar geoengineering (G4NH),and southern hemisphere (SH) solar geoengineering G4SH; **b**, **c**, NH NSAT anomaly and total sea-ice extent ($10^6$ km2); **d**, **e**, SH NSAT anomaly and total sea-ice extent. Sea-ice extents are smoothed by a 10-year simple moving average. Vertical dotted lines at years 2020 and 2070 indicate the start and cessation of solar geoengineering, respectively

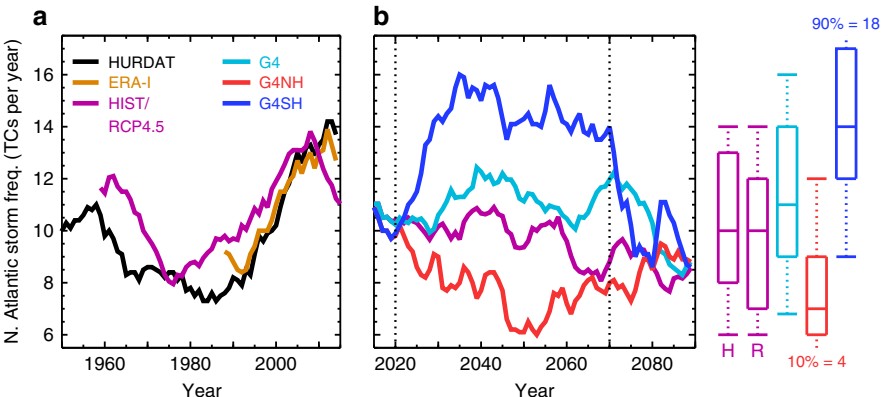

**Fig. 4** Observed and simulated Tropical cyclone frequency. **a** Historical tropical cyclone (TC) frequencies, smoothed by a 10-year simple moving average, for ERA-I[28], the ensemble mean of the HadGEM2-ES HIST simulations and HURDAT2 observations[29]. **b** The same as **a** but for the RCP4.5 and SAI simulations. The box and whisker plots (right) show the 10, 25, 50, 75 and 90% quantiles of the HIST ('H', 1950–2000), RCP4.5 ('R', 2020–2070) and SAI (2020–2070) raw annual TC frequency. G4 refers to a global SAI scenario, G4NH refers to a northern hemisphere SAI scenario, and G4SH refers to a southern hemisphere SAI scenario. Vertical dotted lines at years 2020 and 2070 (**b**) indicate the start and cessation of solar geoengineering, respectively

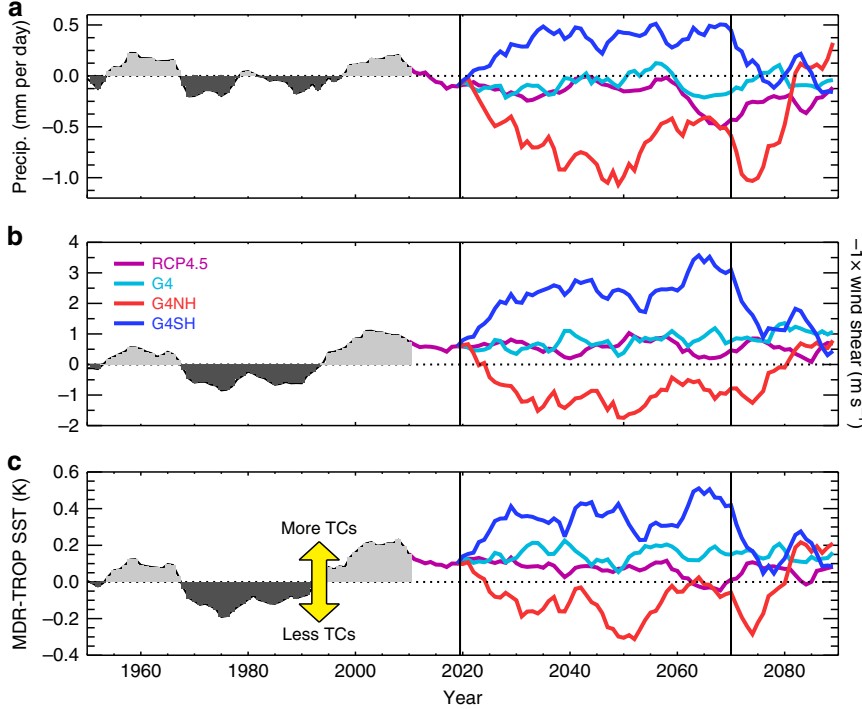

**Fig. 5** Modelled tropical cyclone-related climate indices. **a** June–November (JJASON) precipitation anomaly (relative to 1950–2000) averaged over the hurricane main development region (MDR) (5°–20°N, 15°–85°W). **b** The same as **a** but for inverse vertical zonal-wind shear. **c** The same as **a** but for relative SST. G4 refers to a global SAI scenario, G4NH refers to a northern hemisphere SAI scenario, and G4SH refers to a southern hemisphere SAI scenario. Vertical lines at years 2020 and 2070 indicate the start and cessation of solar geoengineering, respectively

G4NH (Fig. 3b). This result corroborates previous research suggesting that the impacts of SG would not be entirely confined to the perturbed region[4,8,10–12].

**Simulated tropical cyclones.** In the historical period, the model skilfully captures observed TC frequency trends as inferred from TRACK ($r = 0.71$ with HURDAT) including the decline in activity through 1960–1980 and the increase in activity since 1980 (Fig. 4a). The annual-mean TC frequency in HIST is 10.4 (90% confidence intervals (CIs): 5, 16) TCs per year. In RCP4.5, TC frequency decreases between 2020 and 2070 (−0.3 TCs per decade) with an annual-mean frequency of 9.7 (90% CI: 4, 15) TCs per year, while in G4, TC frequency increases slightly relative to HIST (annual-mean = 11.2 (90% CI: 5, 18) TCs per year). The TC frequency changes between 2020 and 2070, and HIST in the RCP4.5 and G4 scenarios are not statistically significant at the 5% level (Supplementary Note 3). G4SH exhibits a marked increase in TC frequency relative to HIST (annual-mean TCs per year = 14.3 (90% CI: 8, 20)), while G4NH conversely exhibits a pronounced reduction (annual-mean TCs per year = 7.6 (90% CI: 2, 13)). The G4NH and G4SH results are consistent with observed TC activity changes following volcanic aerosol enhancements confined to a single hemisphere[16,17]. TC frequency swiftly rebounds to concurrent RCP4.5 levels following the cessation of SAI in G4, G4NH and G4SH in year 2070 (Fig. 4b), which confirms that the SG termination effect[37] extends to North Atlantic TC activity. The TC frequency changes between 2020 and 2070, and HIST in the G4NH and G4SH scenarios are statistically significant at the 5% level (Supplementary Note 3).

**Tropical cyclone proxies.** The progression of AEWs to TCs is contingent on the ambient meteorological conditions, which may act to induce or dissipate the storm. For instance, enhanced wind

shear over the MDR counteracts cyclogenesis[20], whereas a warm ocean surface provides the storm vortex with energy[21]. Historical trends in MDR wind shear, precipitation and relative SST closely correlate with North Atlantic TC activity (Fig. 1 in ref. [13]) and these indices offer an alternative tool to counting vortices for predicting future TC trends. Figure 5 shows various North Atlantic TC indices as extracted from the HadGEM2-ES simulations. It is clear that active (1950–1965, 1995–2014) and inactive (1965–1995) TC periods in the HIST simulation (Fig. 4a) were commensurate with active and inactive periods in the indices (Fig. 5). The same correlations between indices and TC frequency persist in the RCP4.5 and SAI simulations, with G4SH and G4NH exhibiting continuously positive and negative indices, respectively (Fig. 5). This suggests that meteorological conditions presently conducive to cyclogenesis remain conducive in these scenarios. Figure 6 shows maps of precipitation, wind shear, and relative SST anomalies in the G4NH and G4SH scenarios. In G4NH, aerosol-induced cooling of the North Atlantic sea surface (> 30°N) results in a southward shift and strengthening of the African easterly jet (AEJ), enhanced wind shear in the MDR, and anomalous descent and precipitation reduction over the MDR (Fig. 6a–c)[13]. Conversely, preferential cooling of the South Atlantic in G4SH enhances ascent and precipitation in the MDR and shifts the AEJ north, reducing wind shear over the MDR and producing favourable conditions for cyclogenesis (Fig. 6d–f).

Figures 5 and 6 suggest that enhanced TC activity is related to certain climatic conditions in the MDR, in particular enhanced precipitation, attenuated vertical wind shear and a warmer sea surface (relative to the tropical mean). It is important to investigate these relationships using observations and reanalyses to ascertain their practical robustness. Figure 7 shows time series for TC frequency[29], precipitation and vertical wind shear ($U_{850} - U_{25}$)[38], and relative SSTs[39] from reanalyses and observations. From comparing Fig. 7a with Fig. 7b, c, d, it appears that

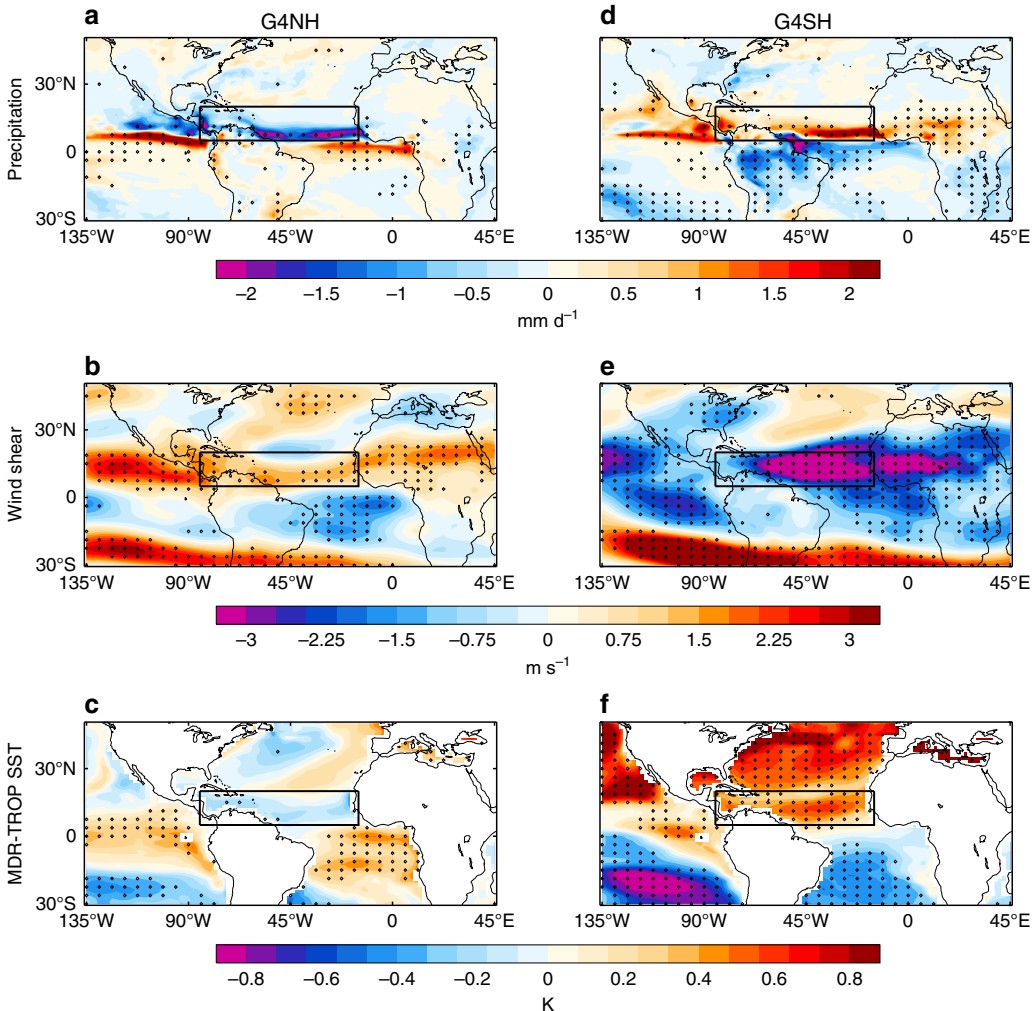

**Fig. 6** Climate anomalies in the hemispheric solar geoengineering simulations. **a** Northern hemisphere only solar geoengineering (G4NH) 2020–2070 JJASON precipitation anomaly relative to 1950–2000. **b** The same as **a** but for vertical zonal-wind shear. **c** The same as **a** but for relative SST. **d**–**f** The same as **a**–**c** but for southern hemisphere only solar geoengineering (G4SH). Stippled regions on the maps show where differences are outside the 90% variability of a 240-year pre-industrial control ensemble mean. The hurricane main development region (MDR, (5°–20°N, 15°–85°W)) is marked by black rectangles

periods of enhanced TC activity in the 20th century coincided with enhanced precipitation and relative SSTs and attenuated vertical wind shear[13], which substantiates our modelling results (Fig. 5). The closest relationship in terms of active and inactive periods in Fig. 7 is between TC activity and relative SST. Statistical models for count data using a Poisson distribution framework can be developed to quantify the observed relationships between TC activity and MDR meteorology (Supplementary Note 4)[21]. Figure 8 shows time series of TC activity from the HadGEM2-ES simulations as determined by applying the statistical relationships from the historical observations (Fig. 7) to the simulated meteorology, where the covariates are anomalies from the 1900 to 2005 mean values. The covariate trends suggest enhanced (attenuated) TC activity in the G4SH (G4NH) simulation between 2020 and 2070 relative to HIST and RCP4.5 (Fig. 8), which substantiates the results of the explicit storm tracking (Fig. 4). We find little evidence to support the hypothesis that the simulated TC frequency changes (Fig. 4) are the result of an El Nino Southern Oscillation response, which further supports the ITCZ-TC connection theory (Supplementary Note 5).

**Statistical-dynamical downscaling.** Statistical-dynamical downscaling models are able to simulate the observed intensity distribution of North Atlantic TCs[40], whereas explicitly simulated

storms are not as intense as those observed (Supplementary Note 6). Therefore, we employ a downscaling model to investigate changes to the most intense storms under global warming and SAI. Forced by HadGEM2-ES meteorology, the model is clearly able to reproduce TC trends in the recent historical period (Fig. 9a), although the frequency of major hurricanes (max windspeed > 96 m s$^{-1}$) is undersimulated in the 1960s compared to HURDAT observations (Fig. 9c). In contrast to the results of the explicit storms (Fig. 4), the model shows a steadily increasing trend in TC frequency in the RCP4.5 scenario over 2020–2070 (Fig. 9a), in agreement with the results of applying downscaling to the CMIP5 ensemble using the RCP8.5 scenario[24]. SAI generally counteracts the intensification of TC activity relative from RCP4.5, except in the interesting case of the first ~10 years in G4SH, which exhibits an increase in major hurricane, hurricane and TC activity (Fig. 9c). G4SH consistently produces the most TCs per year relative to the other SAI scenarios, with 2020–2070 mean frequencies of 12.6, 8.8 and 3.3 TCs, hurricanes and major hurricanes per year, respectively. This can be compared to 10.5, 7.3 and 2.6 for G4NH; 10.8, 7.2 and 2.5 for G4; and 15.3, 11.1 and 4.3 for RCP4.5. The TC frequencies in the SAI simulations are significantly different to RCP4.5, and the G4SH TC frequencies are significantly greater than G4 and G4NH (Supplementary Note 7). Following the cessation of SAI in 2070, TC activity

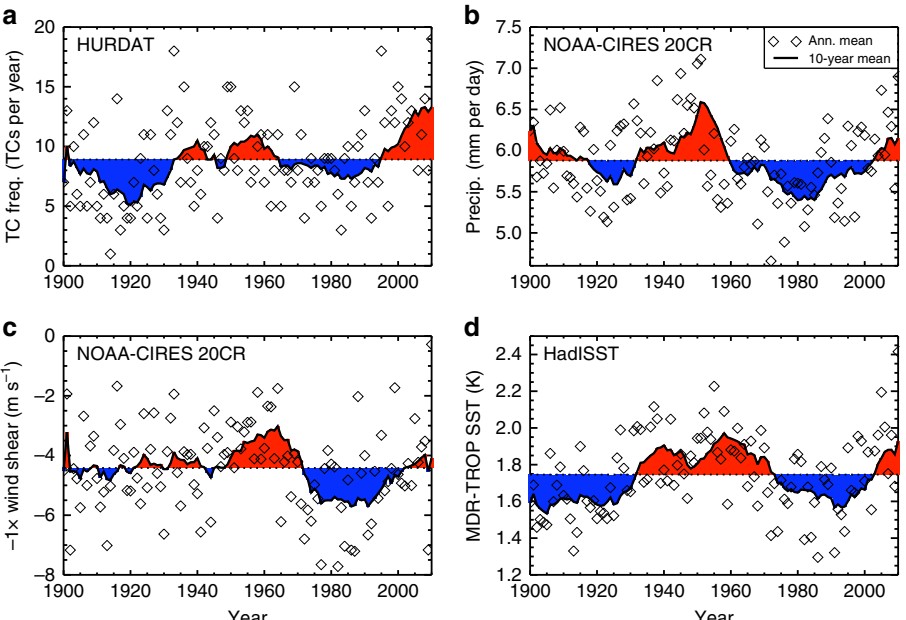

**Fig. 7** Twenty-first century North Atlantic climate indices from observations and reanalyses. June–November (JJASON) observations and reanalysis data time series for: **a**, TC frequency[29]; **b**, hurricane main development region (MDR, (5°–20°N, 15°–85°W)) precipitation and **c**, wind shear[38]; and **d**, sea-surface temperature (MDR minus the Tropics)[39]. Diamonds indicate JJASON average quantities for each year in the MDR or North Atlantic basin, and thick black lines denote 10-year moving averages. Red (blue) regions indicate where moving averages are above (below) the 1900–2010 mean values

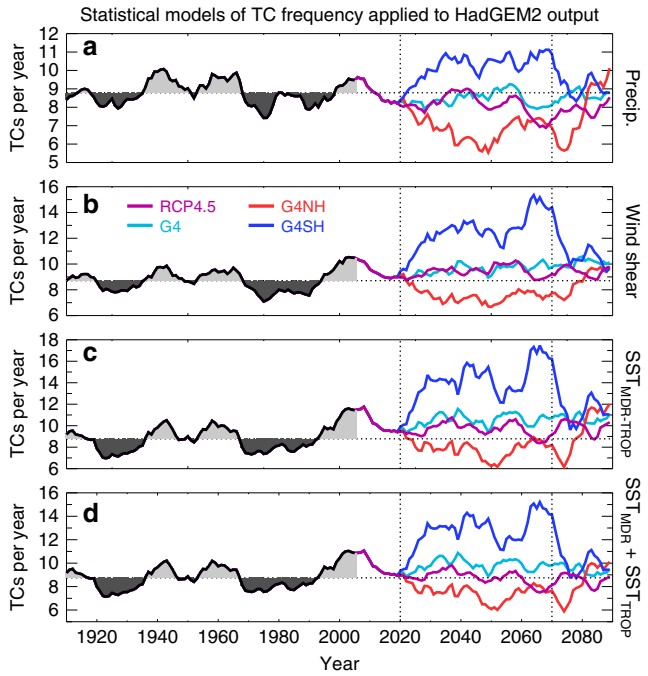

**Fig. 8** Tropical cyclone frequency inferred from various statistical relationships with climate indices. TC frequency inferred from HadGEM2-ES meteorology using statistical relationships developed from historical observations and reanalyses. The covariates used are: **a**, MDR precipitation; **b**, MDR wind shear; **c**, relative sea-surface temperatures (MDR minus tropical mean); and **d**, MDR and tropical SSTs as separate covariates. Vertical dotted lines at years 2020 and 2070 indicate the start and cessation of solar geoengineering, respectively

rebounds to the baseline RCP4.5 activity within ~15 years (Fig. 9a), again confirming that the SG termination effect extends to North Atlantic TCs[31].

## Discussion

A previous multi-model ensemble of GeoMIP G3 and G4 simulations[6] using a temperature-based proxy[41] found that geoengineering reduced hurricane frequency although the statistical significance was marginal. Our results for hemispherically symmetric SAI (G4) are similarly marginal using both the TC-tracking algorithm and three different types of TC proxy (Figs. 4 and 8). However, when employing a statistical-dynamical downscaling algorithm, we find that a global SAI application could reduce TC frequency significantly relative to RCP4.5 (Fig. 9). This disparity between the results of explicit storm modelling from GCM simulations and statistical-dynamical downscaling is not a new result and remains fundamentally unexplained[23,24,42,43]. Nevertheless, there are important commonalities between the results of the explicit storm modelling and statistical-dynamical downscaling. The first is that SAI applied to the SH would increase North Atlantic TC activity relative to a global SAI application (Figs. 4 and 9). The second commonality is that the cessation of SAI would rapidly lead to TC activity rebounding to the base state climate.

The primary result of this research is to demonstrate that single-hemisphere SAI could modulate North Atlantic TC frequency. However, a scenario in which TC frequency is suppressed by NH SAI would unavoidably induce droughts in the Sahel, and vice versa for SH SAI[11,44]. Ideally, our simulations would be replicated by a multi-model ensemble[45]. Because the G4NH and G4SH simulations are unofficial variants of the G4 simulations, they have not been performed by other modelling centres. However, our results are likely to be generally applicable owing to the large body of evidence that if a climate model is forced by cooling one hemisphere, the ITCZ and associated precipitation will migrate towards the opposite hemisphere. This is because the

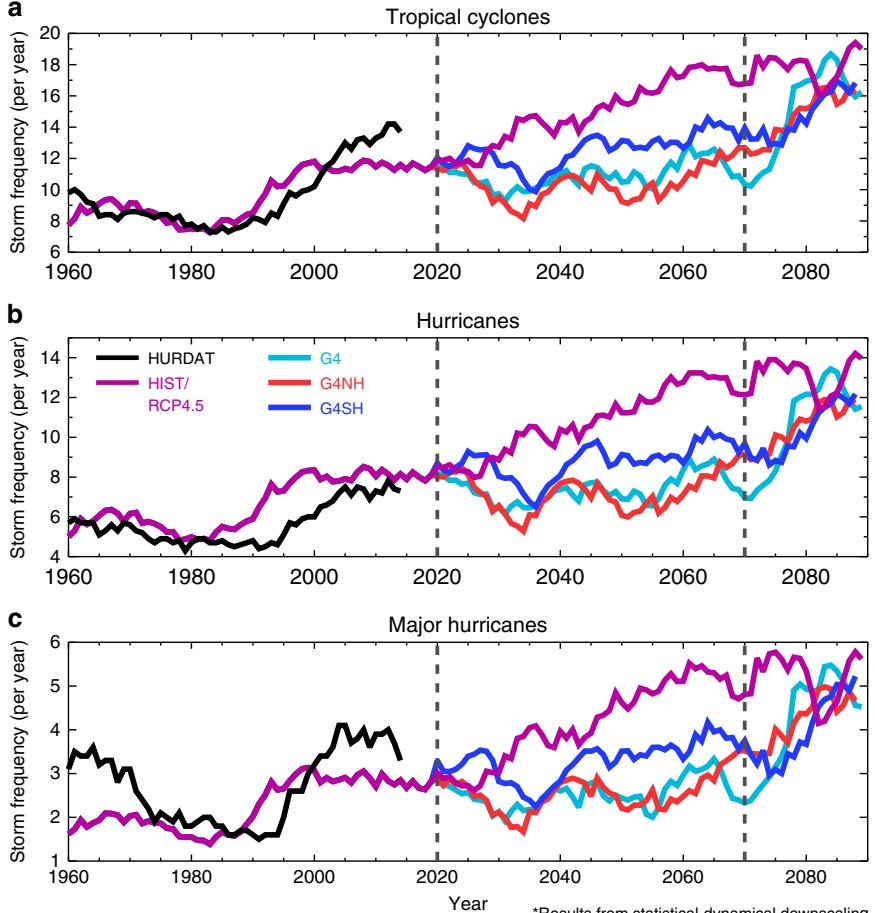

**Fig. 9** Tropical cyclone frequency from downscaling simulations. **a** North Atlantic tropical cyclone (max windspeed >20 m s$^{-1}$) frequency from downscaling simulations using HadGEM2-ES simulated meteorology plotted with observed (HURDAT) TC frequencies. All smoothed with 10-year simple moving averages. **b** The same as **a** but for hurricane (max windspeed > 37 m s$^{-1}$) frequency. **c** The same as **a** and **b** but for major hurricane (max windspeed > 96 m s$^{-1}$) frequency. G4 refers to a global SAI scenario, G4NH refers to a northern hemisphere SAI scenario and G4SH refers to a southern hemisphere SAI scenario. Vertical dashed lines at years 2020 and 2070 indicate the start and cessation of solar geoengineering, respectively

cross-equatorial energy transport adjusts to transport energy away from the warmer hemisphere while the transport of moisture at lower levels in the atmosphere acts in the opposite direction[46–48]. This appears to be a general result that is not dependent on the forcing mechanism indicating that it is the inter-hemispheric cooling gradient across the equator rather than the cooling mechanism that determines the model response[12]. Further, the close relationship between cross-equatorial energy transport in the atmosphere and the ITCZ seen in observations[49] is replicated in GCMs[50], providing confidence in the ability to models to reproduce this behaviour. A shift of the ITCZ to the north in any GCM will lead to an increase in the precipitation in the MDR region, which is a well-documented proxy for hurricane frequency[13]. Thus, although the detailed impacts may differ compared to those presented here, the general conclusions would likely be similar. Further experiments with other models are, however, the only way to substantiate these assertions.

This work reemphasises the perils of unilateral geoengineering, which might prove attractive to individual actors due to a greater controllability of local climate responses, but with inherent additional risk elsewhere[4,8]. The COP21 target of stabilising global-mean warming at 1.5 K above pre-industrial levels[51] appears extremely difficult to achieve even with measures well beyond what would be considered under conventional mitigation scenarios. The overshoot of 1.5 K could theoretically be combated using SAI (Fig. 3), but if applied just to cool the NH, which might

have preferential local climate responses (eg, less Atlantic TCs) for the geoengineering parties, there could be potentially devastating impacts (eg, Sahelian drought) in other regions. We therefore recommend the expeditious implementation of international regulation to control large-scale SG deployment, in order to develop a truly global approach and deter large-scale unilateral deployment.

## Methods

**General circulation model**. HadGEM2-ES is a fully coupled atmosphere–ocean climate model developed by the UK Met Office[52]. The atmospheric sub-model has 38 levels extending to ~40 km, with a horizontal resolution of 1.25° × 1.875° in latitude and longitude, respectively. The model includes the CLASSIC aerosol scheme[53] and an interactive carbon cycle. Briefly, the CLASSIC aerosol scheme was originally designed as a single-moment tropospheric scheme where all major aerosol species are treated as separate external mixtures. Of relevance to this study, is the sulphur scheme that oxidises sulphur dioxide to sulphate aerosol via gas phase oxidation by the hydroxyl radical. Aqueous phase oxidation is of little relevance in the stratosphere owing to the low relative humidities and the absence of clouds. Sulphate aerosol is subsequently removed from the stratosphere via dry deposition into the troposphere in the descending branch of the Brewer–Dobson circulation and tropopause folds. The CLASSIC scheme has been shown to adequately represent simulations of, eg, the eruption of Mount Pinatubo in 1991[32].

**HIST, RCP4.5 and geoengineering simulations**. HadGEM2-ES is forced following the Climate Model Intercomparison Project phase 5 (CMIP5) protocol using historical data from 1860 to 2005 and RCP4.5 scenarios up to 2100[54]. An ensemble of two HIST and three RCP4.5 simulations are performed. Further detail

of the model and simulations is provided in ref. [11]. The G4 scenario follows the GeoMIP protocol[7] and consists of a three member ensemble, while the G4NH and G4SH scenarios use the same total sulphur dioxide ($SO_2$) injection rate (5 Tg [$SO_2$] per year from 2020–2070), but concentrated solely in a single hemisphere (north and south, respectively) and are from single simulations. $SO_2$ is injected evenly between 16 and 25-km altitude (six model levels) in the geoengineering simulations. Figure 2 shows the resultant $SO_4$ 550-nm AOD depth anomalies relative to RCP4.5. It is clear that the geoengineered aerosol is concentrated primarily in the hemisphere of injection, adequately simulating the distribution of aerosol from models with a better resolved stratosphere (Fig. 1). Such hemispherically asymmetric aerosol distributions have been observed subsequent to high-latitude volcanic eruptions such as that of Katmai, which erupted in Alaska in 1913[11].

**Tracking GCM storms.** TC tracking is conducted using the TRACK code (vn. 1.4.7), which has been used for a variety of similar studies[19,55]. For ERA-I, we use full Gaussian resolution (H512 × 256) data sets on 6-h time steps for JJASON 1979–2014[28]. The data is firstly spectrally filtered using spherical harmonic decomposition, which translates the HadGEM-ES and ERA-I data onto a consistent Gaussian grid (128 × 64 longitudes by latitudes) and truncates wavenumbers < 5 and > 42 (ie, T42)[55]. Additionally, we employ a Hoskins filter to smooth the data[56]. Vortices are initially identified by TRACK as local maxima in the 850-hPa relative vorticity field that exceeds a threshold of $0.5 \times 10^{-5}$ s$^{-1}$ at T42 spectral resolution. To identify vortices with a warm core structure (ie, TCs), we reference the tracks to the vorticity field at the 850, 500 and 250-hPa levels at T63 spectral resolution, using a steepest ascent maximisation approach[19]. The criteria used to identify TCs are: (1) that they attain a lifetime ≥ 2 days (ie, $8 \times 6$-h time steps); (2) that cyclogenesis (defined by first identification) must occur between Equator and 30°N; (3) that the maximum T63 intensity of relative vorticity at 850 hPa during the lifetime is ≥ $\xi_1$ for some chosen value of $\xi_1$; (4) that there must be a T63 vorticity maxima at each level up to 250 hPa and the difference in vorticity between 850 and 250 hPa (850–250) ≥ $\xi_V$ for some chosen value of $\xi_V$; (5) that criteria 3 and 4 must be achieved for at least $n$ consecutive 6-h time steps; and finally (6) that they must traverse the North Atlantic hurricane MDR (5°–25°N, 15°–85°W).

For HURDAT2 observations[29], we impose the following criteria: (1) that disturbances must have a lifetime ≥ 2 days, with at least one day within June–November; (2) that the maximum-sustained windspeed exceeds 34 knots; and (3) that they traverse 0°–30°N.

Previous studies have identified North Atlantic TC frequency and intensity as a key low bias in HadGEM2-ES, which has been attributed to the coarse spatial resolution of the model (~200 km at the equator)[12,23]. However, HadGEM2-ES is able to skilfully capture trends in interdecadal TC frequency when forced by historical conditions if TC-intensity thresholds are relaxed[13,57]. For this investigation, we relax the vorticity thresholds in TRACK when applied to the HadGEM2-ES simulations to obtain reasonable fidelity in annual TC frequency compared with ERA-I and HURDAT2. We test different permutations of ($\xi_1$, $\xi_V$, $n$ = 4) for ERA-I and HadGEM2-ES against HURDAT2 to obtain reasonable fidelity in annual-mean TCs (Supplementary Note 8). For ERA-I, we find that (6, 5.5, 4) provides the best fit to HURDAT2. For HadGEM2-ES, we find that (4.5, 3.5, 4) provides the best fit to HURDAT2 (Fig. 4).

**Statistical-dynamical downscaling simulations.** The downscaling technique begins by randomly seeding with weak proto-cyclones the large-scale, time-evolving meteorology of the HadGEM2-ES model. These seed disturbances are assumed to move with the reanalysis-provided large-scale flow in which they are embedded, plus a westward and poleward component owing to planetary curvature and rotation. Their intensity is calculated using the Coupled Hurricane Intensity Prediction System (CHIPS)[58], a simple axis symmetric hurricane model coupled to a reduced upper ocean model to account for the effects of upper ocean mixing of cold water to the surface. Applied to the synthetically generated tracks, this model predicts that a large majority of them dissipate owing to unfavourable environments. Only the 'fittest' storms survive; thus the technique relies on a kind of natural selection. Extensive comparisons to historical events by ref. [40] and subsequent papers provide confidence that the statistical properties of the simulated events are in line with those of historical TCs.

**Code availability.** The code that supports the findings of this study is available from the corresponding author upon reasonable request. The TRACK model is presently available upon request by emailing K. Hodges (k.i.hodges@reading.ac.uk). The CHIPS model is owned by K. Emanuel (emanuel@mit.edu) to whom service requests should be directed.

**Data availability.** The data that support the findings of this study are available from the corresponding author upon reasonable request.

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

## Acknowledgements

A.C.J. was supported by a Met Office/NERC CASE (ref. 580009183) PhD studentship. M. K.H. and J.M.H. were supported by the Natural Environment Research Council/Department for International Development via the Future Climates for Africa (FCFA) funded project 'Improving Model Processes for African Climate' (IMPALA, NE/M017265/1).

## Author contributions

A.C.J. performed the analysis with assistance from J.M.H, K.I.H. and M.K.H. and wrote the paper with assistance from all co-authors. J.M.H., A.J. and N.D. designed the study, and A.J. conducted the simulations. K.E. conducted the statistical downscaling simulations and assisted in the analysis.

## Additional information

**Competing interests:** The authors declare no competing financial interests.

