## [Peer Review File · Nature Communications]

Reviewers' comments:

Reviewer #1 (Remarks to the Author):

This is an interesting paper and worthy of publication. I do have a number of comments and questions. I guess they are mostly relatively minor but the first one perhaps might imply a substantive change to the physical interpretation.

In observational studies of the response to volcanic aerosol, people have found declines in Atlantic TC activity, but it is difficult to separate the response from ENSO since El Niño events have tended to coincide with the big tropical volcanoes. There is some evidence now that in fact volcanoes may cause ENSO events, and in particular that those combined to the NH hemisphere do so (Pausata et al. 2015, PNAS 112, 13784-88). In fig. 4 (esp. 4c) it looks like this is happening here to some extent also. Perhaps the authors could comment on this. The discussion seems to imply that the response is due to the inter-hemispheric gradient in forcing moving the ITCZ but perhaps it is really due to the induced ENSO-like response. These two would have very different implications for the other basins – an El Niño response would tend to increase activity across the north Pacific for example, whereas the inter-hemispheric gradient argument would decrease it. The Atlantic is unrepresentative in that the two effects go the same way there.

It would be good to say a few words (in the supplement is fine) about the aerosol scheme used – I know it is in the reference but please tell us just the basics, a sentence or two about what processes are included etc.

As far as I can tell, it is not stated anywhere what the altitude and vertical structure of the aerosol injection are. Please specify.

About injecting in different hemispheres vs. evenly: my understanding is that because of the stratospheric circulation (briefly mentioned in the text) the more aerosol is put into the tropics, the longer its lifetime, because aerosol injected in the extratropics is flushed out sooner due to the downward advection there. So the most effective strategy to achieve the maximum global mean forcing with the least mass injected would seem to be injection in the tropics, which would tend to distribute aerosol evenly to the two hemispheres since the circulation goes both ways (at least in an annual mean). The authors don't necessarily need to go into this in detail but it seems to me a little more mention of this would be relevant, since they are using a fixed amount of mass and looking at the effects.

Also, by using the same mass for all simulations, the total aerosol burden in the NH or SH is greater (by a factor of 2?) in the single-hemisphere experiments than in the global experiment. This is sort of obvious once one thinks about it but perhaps would be good to spell out clearly.

Are all runs single runs, including the "control" RCP4.5? That is, no ensembles were used? I ask because, for example, when the authors say (lines 91-92) that TC frequency decreases steadily between 2020 and 2070, it doesn't look at all steady to me. Rather it looks pretty hard to be certain that this trend is robust and not a result of aliasing internal variability, i.e. it might be quite different if the run were a little longer or shorter or used different initial conditions. This is not essential to the conclusions but some notion of statistical significance would be helpful. Certainly the adjective "steady" seems misplaced. The differences between the runs look more significant but still some assessment of that formally would be good.

Reviewer #2 (Remarks to the Author):

Review of Solar geoengineering and North Atlantic tropical cyclone frequency by Jones et al. This paper uses results from regional geoengineering experiments using the HadGEM2 earth system model to infer changes in occurrence of Atlantic hurricanes. The authors show that regional stratospheric sulphate aerosol injection in the northern and southern hemispheres alone cause very large changes in cyclone numbers relative to either the control rcp4.5, or the GeoMip G4 experiment of global aerosol injection.

The topic is of wide interest, and the statistical treatment seems convincing for the experiments that did. The paper is concise and assesses much relevant previous research. There do seem to be 2 major problems (points 1 and 2 below), that affect the credibility of the results presented. The authors should also consider how their estimates relate to other studies that have use various proxies for both predicting Atlantic cyclone numbers in future, and which extend the observational record further into the past (points 3 and 4).

Specific Points

1) I think the main difficulty with the work is that it is based on results from only 1 single climate model: HadGEM2-ES. This is a fairly low resolution model, and one that while calculating aerosol formation, does not I think in the model used here, allow aerosol size distribution to change. Comparison recently done of the G4 experiment models (Kashimura et al., 2017) shows some fairly large differences in radiative forcing across the models. These differences suggest that the results using the HADGEM2 model may be significantly different from using a model ensemble. The authors should seek other models to verify the results, or at least discuss how the specifics of the model used may affect results.

2) An issue with the single model of modest resolution when used to calculate extreme cyclones is that the model cannot resolve the detail of the storms. The authors present rather good evidence that using the TRACK software to identify such storms gives credible results compared with observations. But the observational record is relatively short (from 1979-2014, but only 2005 is shown in table S1). The treatment of the 2005-2014 is a bit obscure, its taken as rcp4.5 in the main text, but statistics are not mentioned in tables S1 or S2. While the model and the observations are quite well correlated, this is done by choosing best values for 3 parameters, which are somewhat different from standard values. The authors note that this affects the intensity of the storms tracked, and claim that this is not very important for consideration of cyclone numbers. However, defining the size/number distribution is the most relevant part of future predictions. Much work has shown that under warming cyclone numbers may be decreased, but individual storms may become more intense. While others have suggested that the whole distribution changes, with the most extreme storms relatively most affected by warming. Hence it is not really sufficient to describe how storm frequencies may change without discussing how the intensities may also change.

3) It might be worth trying to compare the model data with observations over the longer period using the reconstructed Atlantic storm surge index that goes to about 1923 (Grinsted et al., 2013), and which also looks at long period and short correlations between TC numbers and various proxies such as MDR temperature anomalies. Comparison with numbers expected using Emanuel's Genesis Potential index may be informative (Emanuel, 2013), or the other methods cited within.

4) The ensemble results for G4 (Moore et al., 2015) are mentioned only in passing, and in a rather incomplete sentence, that does not even mention that the work was about Atlantic hurricane and storm surges. "A study has been performed (Ref. 6) using a multi-model analysis of Geoengineering Model Intercomparison Project (GeoMIP) scenarios G3 and G4 [7], where SAI is applied relatively uniformly to both hemispheres."

The authors' notable relative decrease in cyclone numbers under rcp4.5 compared with G4 (Fig. 2) compared with the reduction in Katrina level storms in Moore et al. (2015) for G4 relative to rcp4.5 should be worthy of mentioning, as this is the only previous study on hurricane impacts of geoengineering. This is also relevant to the other studies of predicting cyclone numbers and size under the CMIP5 and CMIP3 models that have been published, such as Emanuel (2013) and methods he discusses.

References Cited

- Emanuel KA (2013) Downscaling CMIP5 climate models shows increased tropical cyclone activity over the 21st century. *Proc Natl Acad Sci USA* 110(30):12219–12224
- Grinsted A, Moore JC, Jevrejeva S (2012) Homogeneous record of Atlantic hurricane surge threat since 1923. *Proc Natl Acad Sci USA* 109(48):19601–19605.
- Kashimura, H., M. Abe, S. Watanabe, T. Sekiya, D. Ji, J. C. Moore, J.N.S. Cole and B. Kravitz 2017 Shortwave radiative forcing, rapid adjustment, and feedback to the surface by sulfate geoengineering: analysis of the Geoengineering Model Intercomparison Project G4 scenario, *Atmos Chem Phys* 17, 3339-3356, doi: 10.5194/acp-17-3339-2017, 2017.

Reviewer #3 (Remarks to the Author):

This manuscript addresses the influence of solar geoengineering on tropical cyclone frequency in the North Atlantic. I am not aware of any attention directed at this issue, but geoengineering is not my specialty so I could be unaware of similar work. The storyline hangs together nicely, with excellent agreement between simulated and observed TC frequency in the past (even though the model resolution would seem to be insufficient to resolve TC genesis), and a compelling analysis of the mechanisms involved with the interhemispheric influence of stratospheric aerosol on TC frequency. The interhemispheric influence is truly novel, and important contribution, and worthy of publication. Implications for geoengineering strategies are profound, as the paper clearly shows that mitigating one impact of greenhouse gases (polar warming) introduces other serious impacts (suppressed precipitation).

The methodology is sound; although statistical confidence intervals would be helpful, the consistency of the signatures suggests the effects are significant.

Confidence in the mechanisms could be better assessed by determining how much of the past variability in TC frequency can be explained by regressions on the predictors precipitation, wind shear, and SST.

We would like to thank all three of the reviewers for their useful comments on our work. We are pleased that all three reviewers see the merits of the study. We have included detailed responses to each of the comments of the reviewers below, and have incorporated changes based on all of the reviewer's comments in revising the text as suggested.

Reviewer #1 (Remarks to the Author):

This is an interesting paper and worthy of publication. I do have a number of comments and questions. I guess they are mostly relatively minor but the first one perhaps might imply a substantive change to the physical interpretation.

In observational studies of the response to volcanic aerosol, people have found declines in Atlantic TC activity, but it is difficult to separate the response from ENSO since El Nino events have tended to coincide with the big tropical volcanoes. There is some evidence now that in fact volcanoes may cause ENSO events, and in particular that those combined to the NH hemisphere do so (Pausata et al. 2015, PNAS 112, 13784-88). In fig. 4 (esp. 4c) it looks like this is happening here to some extent also. Perhaps the authors could comment on this. The discussion seems to imply that the response is due to the inter-hemispheric gradient in forcing moving the ITCZ but perhaps it is really due to the induced ENSO-like response. These two would have very different implications for the other basins – an El Nino response would tend to increase activity across the north Pacific for example, whereas the inter-hemispheric gradient argument would decrease it. The Atlantic is un-representative in that the two effects go the same way there.

This is an interesting point, worthy of following up. We therefore include a section in the supplementary material (section S4) that assesses the impact of G4/G4NH/GSH on El Nino/La Nina frequency and on the spatial patterns of the induced SST anomalies. Our conclusions are stated below:-

[Line 169, supplement] “Our conclusions from this analysis are that geoengineering the northern/southern hemisphere in HadGEM2-ES simulations has little impact on the commonly used Nino3.4 El Nino/La Nina frequency metric. While SST anomalies under G4NH/G4SH bear some similarity to El Nino/La Nina patterns particularly in the tropics, they are far from identical, particularly in the extra-tropics. Therefore one cannot assert that geoengineering under any of these scenarios would enhance or decrease El Nino/La Nina events, at least from the limited set of results from the HadGEM2-ES model.”

It would be good to say a few words (in the supplement is fine) about the aerosol scheme used – I know it is in the reference but please tell us just the basics, a sentence or two about what processes are included etc.

We agree. As we have a little more space available to us, we include a brief description of the aerosol scheme, with particular emphasis on the sulphur cycle in the main text of the Methods section:-

[Line 226, manuscript] “Briefly, the CLASSIC aerosol scheme was originally designed as a single moment tropospheric scheme where all major aerosol species are treated as separate external mixtures. Of relevance to this study, is the sulphur scheme which oxidises sulphur dioxide to sulphate aerosol via gas phase oxidation by the hydroxyl radical. Aqueous phase oxidation is of little relevance in the stratosphere owing to the low relative humidities and the absence of clouds. Sulphate aerosol is subsequently removed from the stratosphere via dry deposition into the troposphere in the descending branch of the Brewer-Dobson circulation and tropopause folds. The CLASSIC scheme has been shown to adequately represent simulations of e.g. the eruption of Mount Pinatubo in 1991 [Jones et al., 2016b].”

As far as I can tell, it is not stated anywhere what the altitude and vertical structure of the aerosol injection are. Please specify.

We’ve added the following to the Methods section of the main text:-

[Line 245, manuscript] “SO₂ is injected evenly between 16 and 25 km altitude (6 model levels) in the geoengineering simulations.”

About injecting in different hemispheres vs. evenly: my understanding is that because of the stratospheric circulation (briefly mentioned in the text) the more aerosol is put into the tropics, the longer its lifetime, because aerosol injected in the extratropics is flushed out sooner due to the downward advection there. So the most effective strategy to achieve the maximum global mean forcing with the least mass injected would seem to be injection in the tropics, which would tend to distribute aerosol evenly to the two hemispheres since the circulation goes both ways (at least in an annual mean). The authors don’t necessarily need to go into this in detail but it seems to me a little more mention of this would be relevant, since they are using a fixed amount of mass and looking at the effects.

Agreed. We now include a new section in the supplementary material (section S2) that describes the injection strategy with an atmosphere only version of HadGEM2 with a better representation of the stratospheric circulation owing to increasing the height of the highest model layer. It also provides an assessment of what altitudes and latitudes of injection lead to hemispherically asymmetric stratospheric aerosol distributions, and what injection strategies lead to an even distribution between the hemispheres.

The reviewer is also correct on a further point – injection in the tropics (15N-15S) can lead to an even distribution provided the injection is high enough. However, outside of the tropics and at somewhat lower altitudes the aerosol will tend to be confined to the hemisphere of

injection. HadGEM2-ES is able to represent the hemispherically asymmetric distribution that one might expect from 'sub-optimal' geoengineering.

Also, by using the same mass for all simulations, the total aerosol burden in the NH or SH is greater (by a factor of 2?) in the single-hemisphere experiments than in the global experiment. This is sort of obvious once one thinks about it but perhaps would be good to spell out clearly.

This detail was included in the supplementary material, but may be more appropriate in the methods section. The methods section now includes i) General Circulation model including the aerosol scheme with focus on the stratospheric aerosol scheme ii) HIST, RCP and Geoengineering simulations, which describe the historical simulations, the RCP4.5 simulations and the G4, G4NH and G4SH simulations., iii) methodology for the tracking of GCM storms, iv) methodology for statistical dynamical downscaling simulations. The statistical-dynamical downscaling simulations were performed to assess changes to storm intensity under solar geoengineering.

Are all runs single runs, including the "control" RCP4.5? That is, no ensembles were used? I ask because, for example, when the authors say (lines 91-92) that TC frequency decreases steadily between 2020 and 2070, it doesn't look at all steady to me. Rather it looks pretty hard to be certain that this trend is robust and not a result of aliasing internal variability, i.e. it might be quite different if the run were a little longer or shorter or used different initial conditions. This is not essential to the conclusions but some notion of statistical significance would be helpful. Certainly the adjective "steady" seems misplaced. The differences between the runs look more significant but still some assessment of that formally would be good.

We now describe the number of ensembles used in the simulations in the revised methods section. The fact that we use an ensemble of three simulations for the RCP4.5 and HIST simulations provides confidence that we are not simply observing variability from a single member simulation. We agree that the TC frequency in the ensemble-mean of RCP4.5 (Fig. 2b) does not decrease 'steadily' and we've removed that word from the text. In section S6 of the supplementary material, we assess the significance of the TC changes using a Wilcoxon rank-sum test (WRST) and a Student's t-test. We find that the G4NH and G4SH changes are significant at the 5 % level for the period 2020-2070 relative to 1950-2000, whereas the corresponding TC changes in RCP4.5 and G4 are not significant (Supplementary Table S6.1). We have now made this point clear in the main text:

[Line 114, manuscript] "The TC frequency changes between 2020-2070 in the RCP4.5 and G4 scenarios and HIST are not statistically significant at the 5 % level (Supplementary Table S6.1)."

Reviewer #2 (Remarks to the Author):

Review of Solar geoengineering and North Atlantic tropical cyclone frequency by Jones et al. This paper uses results from regional geoengineering experiments using the HadGEM2 earth

system model to infer changes in occurrence of Atlantic hurricanes. The authors show that regional stratospheric sulphate aerosol injection in the northern and southern hemispheres alone cause very large changes in cyclone numbers relative to either the control rcp4.5, or the GeoMip G4 experiment of global aerosol injection.

The topic is of wide interest, and the statistical treatment seems convincing for the experiments that did. The paper is concise and assesses much relevant previous research. There do seem to be 2 major problems (points 1 and 2 below), that affect the credibility of the results presented. The authors should also consider how their estimates relate to other studies that have use various proxies for both predicting Atlantic cyclone numbers in future, and which extend the observational record further into the past (points 3 and 4).

Specific Points

1) I think the main difficulty with the work is that it is based on results from only 1 single climate model: HadGEM2-ES. This is a fairly low resolution model, and one that while calculating aerosol formation, does not I think in the model used here, allow aerosol size distribution to change. Comparison recently done of the G4 experiment models (Kashimura et al., 2017) shows some fairly large differences in radiative forcing across the models. These differences suggest that the results using the HadGEM2 model may be significantly different from using a model ensemble. The authors should seek other models to verify the results, or at least discuss how the specifics of the model used may affect results.

While the reviewer is correct that there may be differences in the details between the response of HadGEM2-ES and other models, it is simply not possible to run a multi-model ensemble with other models without getting agreement from other modelling centres. This is because the G4NH and G4SH variants are not standard simulations of GeoMIP. G4NH and G4SH were originally designed by the authors to assess the impact of ‘unilateral’ geoengineering on a number of phenomena such as the impact on Sahelian rainfall (Haywood et al., 2013). We note that the work by Haywood et al. (2013) was with a single model and therefore there is some precedence for publishing such papers in the Nature group of journals and the paper is well cited (84 total, average 16.8 times per year; web of science).

The reviewer suggests that we should at least discuss how the specifics of the model may impact the results. We therefore provide the following text:-

[Line 188, manuscript] “Ideally, our simulations would be replicated by a multi-model ensemble [Kravitz *et al.*, 2013]. Because the G4NH and G4SH simulations are unofficial variants of the G4 simulations they have not been performed by other modelling centres. However, our results are likely to be generally applicable owing to the large body of evidence that if a climate model is forced by cooling one hemisphere, the ITCZ and associated precipitation will migrate towards the opposite hemisphere. This is because the cross equatorial energy transport adjusts to transport energy away from the warmer hemisphere while the transport of moisture at lower levels in the atmosphere acts in the opposite direction [e.g. Hwang and Frierson, 2013; Stephens *et al.*, 2016, Hawcroft *et al.*, 2016]. This appears to be a general result that is not dependent on the forcing mechanism indicating that it is the inter-hemispheric cooling gradient across the equator rather than the cooling mechanism that

determines the model response [e.g. Haywood *et al.*, 2016]. Further, the close relationship between cross-equatorial energy transport in the atmosphere and the ITCZ seen in observations [e.g. Schneider *et al.*, 2014] is replicated in GCMs [e.g. Loeb *et al.*, 2015, Hawcroft *et al.*, in review], providing confidence in the ability of models to reproduce this behaviour. A shift of the ITCZ to the north in any GCM will lead to an increase in the precipitation in the MDR region, which is a well documented proxy for hurricane frequency¹³. Thus, although the detailed impacts may differ compared to those presented here, the general conclusions would likely be similar. Further experiments with other models are, however, the only way to substantiate this assertion.”

References:

Kravitz, B, A. Robock, P.M. Forster, J.M. Haywood, M.G. Lawrence, and H. Schmidt (2013), An overview of the Geoengineering Model Intercomparison Project (GeoMIP), *J. Geophys.Res.Atmos.*, 118, 13,103–13,107, doi:10.1002/2013JD020569.

Hwang, Y. T., and D. M. Frierson (2013), Link between the double-intertropical convergence zone problem and cloud biases over the Southern Ocean, *Proc. Natl. Acad. Sci. U.S.A.*, 110(13), 4935–4940.

Schneider T., *et al.* (2014), Migrations and dynamics of the intertropical convergence zone. *Nature*, 513(7516):45–53

Loeb N. G., *et al.* (2015), Observational constraints on atmospheric and oceanic cross-equatorial heat transports: revisiting the precipitation asymmetry problem in climate models. *Clim. Dyn.* 46, 9-10, 3239–3257

Stephens G. L., *et al.* (2016), The curious nature of the hemispheric symmetry of the earth's water and energy balances. *Curr. Clim. Chang. Rep.* , 2(4):135–147

Hawcroft, M. K., *et al.*, in review. The contrasting climate response to tropical and extratropical energy perturbations, in review with *Climate Dynamics*.

2) An issue with the single model of modest resolution when used to calculate extreme cyclones is that the model cannot resolve the detail of the storms. The authors present rather good evidence that using the TRACK software to identify such storms gives credible results compared with observations. But the observational record is relatively short (from 1979-2014, but only 2005 is shown in table S1). The treatment of the 2005-2014 is a bit obscure, its taken as rcp4.5 in the main text, but statistics are not mentioned in tables S1 or S2. While the model and the observations are quite well correlated, this is done by choosing best values for 3 parameters, which are somewhat different from standard values. The authors note that this affects the intensity of the storms tracked, and claim that this is not very important for consideration of cyclone numbers. However, defining the size/number distribution is the most relevant part of future predictions. Much work has shown that under warming cyclone numbers may be decreased, but individual storms may become more intense. While others

have suggested that the whole distribution changes, with the most extreme storms relatively most affected by warming. Hence it is not really sufficient to describe how storm frequencies may change without discussing how the intensities may also change.

These are good points and have motivated us to extend our study to investigating the response of storm intensity changes to SAI. Firstly, we include a section in the supplementary material (section S8) in which we explore changes to the intensity distribution between HIST and the SAI/RCP4.5 simulations. We use the maximum-sustained wind speed as our intensity metric and assess whether the frequency of the most intense storms changes (Supplementary Figure S8.3). We find that the frequency of intense storms increases in RCP4.5 while there are fewer weaker storms in that scenario (Fig S8.3), which is consistent with results from other GCMs [e.g. Walsh *et al.*, 2015]. Interestingly, all of the SAI scenarios are unable to offset the increase in the most intense storms, which could be due to the aerosol injection scenario utilised in these simulations. We note that the model is unable to reproduce the TC intensity distribution from observations or the reanalysis (Supplementary Figure S8.1) which reduces our confidence in the robustness of the intensity changes.

The most significant change to the manuscript is the inclusion of statistical-dynamical downscaling simulations using the HadGEM2-ES meteorology. We decided to extend our study using downscaling techniques in order to explicitly assess changes to storm intensity in order to respond fully to the issues the reviewer has raised (Fig. 5 has been added to the manuscript). We describe the downscaling simulations in the Methods section of the manuscript and briefly in the main text. Interestingly, the statistical-dynamical downscaling simulations produce different results in terms of TC frequency projections from the explicitly simulated storms (comparing Fig. 2 with Fig. 5). This is especially true for the RCP4.5 scenario, which exhibits an increasing trend in TC activity over the 2020-2070 time period in the downscaling simulations. SAI appears to counteract this increase in TC activity in the SAI scenarios (Fig. 5). This disparity between the results of Emanuel *et al* (2006)'s downscaling model and explicit storms from GCMs is not a new result [Emanuel, 2013], which we emphasize in the Discussion section of the manuscript. However, a common result from the two TC identification methods is that G4SH consistently produces more storms in the North Atlantic basin than a G4 or G4NH (Fig. 5). This corroborates the most important result from this study – that regional SAI could effectively modulate TC frequency. Additionally, incorporating the statistical-dynamical downscaling results underlines an important disparity between current TC identification methods that remains unresolved.

References:

- Emanuel, K., *et al.* (2006), A statistical deterministic approach to hurricane risk assessment, *Bull. Am. Meteorol. Soc.*, 87, 299–314
- Emanuel, K. A. (2013), Downscaling CMIP5 climate models shows increased tropical cyclone activity over the 21st century, *Proc. Natl. Acad. Sci.*, 110(30), 12,219–12,224.
- Walsh, K. J. E., *et al.* (2015), Hurricanes and climate: The U.S. CLIVAR working group on hurricanes, *Bull. Am. Meteorol. Soc.*, 96(6), 997–1017

3) It might be worth trying to compare the model data with observations over the longer period using the reconstructed Atlantic storm surge index that goes to about 1923 (Grinsted et al., 2013), and which also looks at long period and short correlations between TC numbers and various proxies such as MDR temperature anomalies. Comparison with numbers expected using Emanuel's Genesis Potential index may be informative (Emanuel, 2013), or the other methods cited within.

The historical simulations only saved 6 hourly data for the period 1950-2005 so we are unable to track storms any further back than 1950. The TC proxies (precip, SSTs) have already been analysed by Dunstone *et al.* (2013) (their Fig. 1) using HURDAT observations as far back as 1880. They found that observed storm activity trends in the North Atlantic basin were well correlated with observed TC proxies in both the observations and the HadGEM2-ES simulations

Dunstone, N. J., *et al* (2013), Anthropogenic aerosol forcing of Atlantic tropical storms, *Nature Geosci.*, 6, 534–539

4) The ensemble results for G4 (Moore et al., 2015) are mentioned only in passing, and in a rather incomplete sentence, that does not even mention that the work was about Atlantic hurricane and storm surges. “A study has been performed (Ref. 6) using a multi-model analysis of Geoengineering Model Intercomparison Project (GeoMIP) scenarios G3 and G4 [7], where SAI is applied relatively uniformly to both hemispheres.”

We agree and we have improved this sentence as suggested. We also now refer more explicitly to the similarities/differences to the study of Moore et al (2015) as suggested below. An improved statistical analysis is also included in which we have tested the significance of the storm activity changes using a Wilcoxon rank sum test (Supplementary Section S6)

The authors' notable relative decrease in cyclone numbers under rcp4.5 compared with G4 (Fig. 2) compared with the reduction in Katrina level storms in Moore et al. (2015) for G4 relative to rcp4.5 should be worthy of mentioning, as this is the only previous study on hurricane impacts of geoengineering. This is also relevant to the other studies of predicting cyclone numbers and size under the CMIP5 and CMIP3 models that have been published, such as Emanuel (2013) and methods he discusses.

We have extended the Discussion to include a direct comparison of our results with Moore et al. (2015) and with other model-based studies:-

[Line 170, manuscript] “A previous multi-model ensemble of GeoMIP G3 and G4 simulations (Moore et al, 2015) using a temperature based proxy (Grinsted et al., 2013) found that geoengineering reduced TC frequency although the statistical significance was marginal when compared to RCP4.5. Our results for hemispherically symmetric SAI (G4) are similarly marginal using both the TC tracking algorithm and three different

types of TC proxy (Figs 2 &3). However, when employing a statistical-dynamical downscaling algorithm, we find that a global SAI application could reduce TC frequency significantly relative to RCP4.5 (Fig. 5). This disparity between the results of explicit storm modelling from GCM simulations and statistical-dynamical downscaling is not a new result and remains fundamentally unexplained [Emanuel, 2013; Camargo, 2013; Murakami *et al.*, 2014; Walsh *et al.*, 2015]. Nevertheless, there are important commonalities between the results of the explicit storm modelling and statistical-dynamical downscaling. The first is that SAI applied to the southern hemisphere would increase North Atlantic TC activity relative to a global SAI application (Figs 2 &5). The second commonality is that the cessation of SAI would rapidly lead to TC activity rebounding to the base state climate.”

References:

- Emanuel, K. A. (2013), Downscaling CMIP5 climate models shows increased tropical cyclone activity over the 21st century, *Proc. Natl. Acad. Sci.*, 110(30), 12,219–12,224.
- Camargo, S. J. (2013), Global and regional aspects of tropical cyclone activity in the CMIP5 models. *J. Clim.*, 26, 9880–9902
- Murakami, H., *et al.* (2014), Influences of model biases on projected future changes in tropical cyclone frequency of occurrence. *J. Clim.*, 27, 2159–2181,
- Walsh, K. J. E., *et al.* (2015), Hurricanes and climate: The U.S. CLIVAR working group on hurricanes, *Bull. Am. Meteorol. Soc.*, 96(6), 997–1017

References Cited

- Emanuel KA (2013) Downscaling CMIP5 climate models shows increased tropical cyclone activity over the 21st century. *Proc Natl Acad Sci USA* 110(30):12219–12224
- Grinsted A, Moore JC, Jevrejeva S (2012) Homogeneous record of Atlantic hurricane surge threat since 1923. *Proc Natl Acad Sci USA* 109(48):19601–19605.
- Kashimura, H., M. Abe, S. Watanabe, T. Sekiya, D. Ji, J. C. Moore, J.N.S. Cole and B. Kravitz 2017 Shortwave radiative forcing, rapid adjustment, and feedback to the surface by sulfate geoengineering: analysis of the Geoengineering Model Intercomparison Project G4 scenario, *Atmos Chem Phys* 17, 3339-3356, doi:10.5194/acp-17-3339-2017, 2017.

We now include reference to the Grinsted *et al.* (2012) study in reference to the model used by Moore *et al.* (2015) [Line 171] and the Emanuel (2013) study [various lines].

Reviewer #3 (Remarks to the Author):

This manuscript addresses the influence of solar geoengineering on tropical cyclone frequency in the North Atlantic. I am not aware of any attention directed at this issue, but geoengineering is not my specialty so I could be unaware of similar work. The storyline

hangs together nicely, with excellent agreement between simulated and observed TC frequency in the past (even though the model resolution would seem to be insufficient to resolve TC genesis), and a compelling analysis of the mechanisms involved with the interhemispheric influence of stratospheric aerosol on TC frequency. The interhemispheric influence is truly novel, and important contribution, and worthy of publication. Implications for geoengineering strategies are profound, as the paper clearly shows that mitigating one impact of greenhouse gases (polar warming) introduces other serious impacts (suppressed precipitation).

The methodology is sound; although statistical confidence intervals would be helpful, the consistency of the signatures suggests the effects are significant.

We agree that a more detailed statistical analysis of confidence in the results is warranted, and we've now included statistical testing in our analysis. We perform Wilcoxon rank sum tests (WRSTs) and t-tests to assess whether TC frequency changes are significant (Supplementary Sections S6 & S9). As a result, we now include 5% and 95% ranges on the TC frequencies in the text (Subsection 'Simulated TCs' in the Results section) and have highlighted in the text that the TC changes in the RCP4.5 and G4 scenarios are not significant when comparing 2020-2070 with HIST (Supplementary table S6.1).

Confidence in the mechanisms could be better assessed by determining how much of the past variability in TC frequency can be explained by regressions on the predictors precipitation, wind shear, and SST.

We agree with the reviewer and have added Section 7 to the supplementary material in which we explore statistical relationships between the covariates and TC frequency using reanalysis and observational data. We have also made reference to Section S6 in the main text.

REVIEWERS' COMMENTS:

Reviewer #1 (Remarks to the Author):

The authors have addressed my concerns well enough and in my view the manuscript is acceptable for publication.

Reviewer #2 (Remarks to the Author):

Review of Solar geoengineering and North Atlantic tropical cyclone frequency by Jones et al. This paper uses results from regional geoengineering experiments using the HadGEM2 earth system model to infer changes in occurrence of Atlantic hurricanes. The authors show that regional stratospheric sulphate aerosol injection in the northern and southern hemispheres alone cause very large changes in cyclone numbers relative to either the control rcp4.5, or the GeoMip G4 experiment of global aerosol injection.

The topic is of wide interest, and the statistical treatment seems convincing for the experiments that did. The paper is concise and assesses much relevant previous research.

The revised version answers all the comments I had on the original submission. Furthermore it provides an extremely comprehensive review of possible confounding factors, and strengthens the conclusions using a statistical downscaling method.

John Moore